# Combination Therapy with Nusinersen and Onasemnogene Abeparvovec-xioi in Spinal Muscular Atrophy Type I

**DOI:** 10.3390/jcm10235540

**Published:** 2021-11-26

**Authors:** Andrada Mirea, Elena-Silvia Shelby, Mihaela Axente, Mihaela Badina, Liliana Padure, Madalina Leanca, Vlad Dima, Corina Sporea

**Affiliations:** 1Faculty of Midwifery and Nursing, University of Medicine and Pharmacy “Carol Davila”, 37 Dionisie Lupu Street, 020021 Bucharest, Romania; mihaela.axente@drd.umfcd.ro (M.A.); mihaela.badina@drd.umfcd.ro (M.B.); corina.sporea@gmail.com (C.S.); 2Scientific Research Nucleus, National University Center for Children Neurorehabilitation “Dr. Nicolae Robanescu”, 44 Dumitru Minca Street, 041408 Bucharest, Romania; silviajdx@yahoo.com (E.-S.S.); lilianapadure@gmail.com (L.P.); mada_mada332@yahoo.com (M.L.); 3Clinical Hospital of Obstetrics and Gynecology “Filantropia”, 11 Ion Mihalache Avenue, 011132 Bucharest, Romania; dima.vlad@yahoo.com

**Keywords:** spinal muscular atrophy, combined modifying therapy, nusinersen, onasemnogene abeparvovec-xioi, early treatment, motor evolution, ventilation improvement

## Abstract

Background: Spinal muscular atrophy (SMA) is a neuromuscular progressive disease, characterized by decreased amounts of survival motor neuron (SMN) protein, due to an autosomal recessive genetic defect. Despite recent research, there is still no cure. Nusinersen, an antisense oligonucleotide acting on the *SMN2* gene, is intrathecally administered all life long, while onasemnogene abeparvovec-xioi, a gene therapy, is administered intravenously only once. Both therapies have proven efficacy, with best outcomes obtained when administered presymptomatically. In recent years, disease-modifying therapies such as nusinersen and onasemnogene abeparvovec-xioi have changed the natural history of SMA. Methods: We observed seven SMA type I patients, who received both therapies. We compared their motor function trajectories, ventilation hours and cough assist sessions to a control group of patients who received one therapy, in order to investigate whether combination therapy may be more effective than a single intervention alone. Results: Patients who received both therapies, compared to the monotherapy cohort, had the same motor function trajectory. Moreover, it was observed that the evolution of motor function was better in the 6 months following the first therapy than in the first 6 months after adding the second treatment. Conclusions: Our results suggest that early treatment is more important than combined therapy.

## 1. Introduction

Spinal muscular atrophy (SMA) is a neuromuscular genetic disease with a broad spectrum of symptoms characterized by progressive loss of the motor neurons located mainly in the anterior horns of the spinal cord, with subsequent progressive muscle weakness and atrophy [1,2,3]. About 95% of the cases are a consequence of decreased amounts of survival motor neuron (SMN) protein, due to deletions or mutations in the *SMN1* gene, located on chromosome 5q (5q13.2) (5q SMA) [1].

Globally, SMA is the second highest genetic cause of infant mortality after cystic fibrosis [4].

About 5% of the total SMA cases, caused by 16 genes other than SMN1, are encompassed within the name of non-5q SMA [5].

SMA has an incidence of about 1 in 10,000 live births [6] and a worldwide carrier frequency of 1 in 40 to 1 in 60 [7]. The main symptoms are bilateral, symmetrical weakness and progressive muscle atrophy [2], with predominantly proximal onset and a more significant lower-limb involvement compared to the upper limbs. Deep tendon reflexes are either absent or markedly decreased [8].

The *SMN1* gene produces about 90% of the total quantity of bodily SMN protein, the rest being encoded by the *SMN2* gene [9]. The difference between the *SMN1* and *SMN2* genes is a C to T transition in SMN2′s exon 7, which leads to the removal, following splicing, of exon 7 from the SMN2 pre-messenger RNA [10,11], leading to the synthesis of a nonfunctional SMN protein that is subsequently degraded in the ubiquitin–proteasomal pathway [9]. Several studies have shown that the increased number of SMN2 copies modulates the severity and manifestations of 5q-SMA [12,13,14].

According to clinical presentation and severity, 5q-SMA can be classified into five types, annotated from type 0—the most severe (fetal period onset) to type 4—the least severe (adult onset) [7].

Type 0 [15] is the rarest form of 5q-SMA, occurring in less than 1% of all cases. Onset of symptoms is frequently before birth, manifesting as an absence of fetal movements. Patients are born with severe hypotonia and muscle soreness, facial diplegia, contractures, generalized absence of deep tendon reflexes, dysphagia and respiratory failure. A large number of these patients also present with congenital heart defects. Death occurs in the first weeks of life, before the age of 6 months [16,17].

Type 1 SMA, also called Werdnig–Hoffmann disease, is the most frequent type of 5q-SMA, accounting for about 55% of all cases. Patients present hypotonia (“floppy infant”), predominantly muscular weakness [18], weak cry, shortness of breath and abdominal breathing, difficulty sucking and swallowing as well as tongue fasciculations. Unassisted sitting position is never acquired and 75% of the patients die or require permanent ventilation by the age of 13.6 months [19]. Thus, in the absence of treatment and standards of care, death occurs in the first two years of life due to respiratory failure [20].

Type 2 SMA (Dubowitz disease) encompasses about 20% of all cases. Onset is after 6 months of age (between 6 and 18 months) [15]. Patients are able to sit but cannot maintain the standing position without support or independent ambulation [16]. These patients can also have difficulties swallowing, tongue fasciculations and respiratory failure. Usually, the facial and ocular muscles of these patients are not affected. Life expectancy is longer, as these patients can survive without treatment until adulthood [7].

Type 3 SMA, also known as Kugelberg–Welander disease, occurs after the age of 18 months [21]. Some patients lose ambulation during childhood, while in others ambulation is preserved until adulthood. All patients acquire gait, but their proximal muscle deficit leads to frequent falls and difficulty climbing stairs, with gradual loss of ambulation [7].

Onset of type 4 SMA occurs during adulthood, usually after the first 2 decades of life. Mobility is maintained throughout the entire life and the life expectancy is normal [7,15].

As revealed previously, prognosis of SMA was very severe before the approval of disease-modifying therapies. Currently, three promising therapies have been approved by the U.S. Food and Drug Administration (FDA) [22] and the European Medicines Agency (EMA) [23]. These therapies consist of either targeting of *SMN2* gene pre-m-RNA (nusinersen [24] and risdiplam [25]) or gene therapy (GT) using an AAV9-modified functional *SMN1* gene [26].

Nusinersen [27] is the first approved SMN2 pre-mRNA targeted therapy for SMA (December 2016 for United States of America and May 2017 for Europe) [28,29]. It consists of modified antisense oligonucleotides [30] designed to bind to the intronic splice silencer site located in intron 7 of SMN-2 pre-messenger RNA. This, in turn, promotes exon 7 inclusion at the SMN2 messenger RNA level [27]. Thus, a complete and functional SMN protein is obtained. Nusinersen is administered to patients of all ages, by intrathecal administration, the first four loading doses being given in the first two months, then one injection every four months [31,32].

Nusinersen proved highly efficient in improving motor function [33,34]. Several studies have shown that nusinersen-treated patients had variable respiratory [35] and nutritional status improvements. Hospitalization time was reduced with no significant adverse reactions post-administration. Post-administration adverse events occur after lumbar punctures (i.e., headache, vomiting, fever, low back pain, etc.) [32,36] at the usual frequency (10–20%).

Onasemnogene abeparvovec-xioi (with the commercial name Zolgensma) is a gene replacement therapy (GRT) that has been approved for treatment of SMA patients under the age of 2 years (according to the FDA) or a maximum weight of 21 kg and no age limit (according to the EMA). Therapy consists of the administration of the SMN1 transgene, using, as a vector, adeno-associated virus 9 (AAV-9). The administration is conducted by a single intravenous infusion. This treatment restores SMN protein in motor neurons, improves motor function and increases life expectancy in type I patients, with the effects improving and persisting even 6 years after the first administration [37]. The most common adverse effects of Zolgensma were increased liver enzymes [38], fever, vomiting and decreased platelet levels. Less commonly, hemolytic anemia was reported, leading to severe complications, including death in a few cases [39].

The drug was approved by the FDA in May 2019 [40] and obtained conditional approval from the European Medicines Agency [41] for the treatment of patients weighing up to 21 kg [42].

In January 2020, the Managed Access Program (MAP) was launched, aiming to distribute 100 doses [43] by the end of 2020 to countries around the world where the use of Zolgensma had not yet been approved. Only patients under 2 years of age were included, according to the FDA approval. Recently, this program has been renewed, with another 100 doses planned for 2021 [44].

Currently, in Romania, the cost of nusinersen is covered by the National Health Insurance House through the National Plan for Rare Diseases [45].

Following FDA approval and the conditional authorization by the EMA of Zolgensma, there were a few cases of pediatric patients who received both types of therapy concurrently. As is already known, the two drugs have different mechanisms of action, but the long-term effects of their concurrent administration are not known yet. According to a study from 2020 on only five patients, combination therapy with nusinersen and Zolgensma was well tolerated [46].

## 2. Materials and Methods

In this retrospective observational single center study, we evaluated safety and treatment effects of combining disease-modifying therapies, such as nusinersen and onasemnogene abeparvovec-xioi, in SMA type I patients.

SMA diagnoses and type classification were made by the pediatric neurologists at our institute (M.A. and M.L.). Genetic diagnosis was conducted using the Multiplex Ligation-Dependent Probe Amplification (MLPA) technique [47]. *SMN1* gene exons 7/8 homozygous deletion was confirmed in all reported patients. The number of SMN2 copies was also determined by MLPA (all included patients had 2 SMN2 copies).

All patients were treated in the National University Center for Pediatric Neurorehabilitation “Dr. Nicolae Robanescu” (NUCCNNR) in Bucharest, Romania, between July 2019 and May 2021, and we applied to all of them standards of care according to international guidelines [20,21,48], in order to provide uniformity of care among the patients.

We designed this observational study after obtaining ethical committee approval at our hospital; data were collected during periodic evaluations, according to our protocols for drug administration, and all patients’ parents signed informed consents.

Patients were monitored for disease progression and treatment efficacy, using the Children’s Hospital of Philadelphia Infant Test of Neuromuscular Disorders (CHOP-INTEND) [49,50]. The test was performed at the initiation of each therapy and then every 6 months by trained physical therapists from the Neuromuscular Diseases Department in our institute.

We also evaluated the need for non-invasive ventilation (NIV) [51] and also for cough assist sessions [51,52], noting the daily required number of hours/sessions for NIV and for cough assist.

We created 2 groups. In the first one, we included 7 patients (4 males and 3 females) with SMA type I, aged (at the initiation of the first treatment) between 2 months and 6 months old, who received combination therapy as follows: six of them received nusinersen first, as this drug is reimbursed in Romania, then onasemnogene abeparvovec-xioi, which had been accessed through the early-access MAP or crowdfunding. Only one of them received onasemnogene abeparvovec-xioi (through a trial in France) first and then nusinersen 18 months later.

In the second (control) group, we selected 6 other patients who received only nusinersen. Thus, we included 6 patients (3 males and 3 females aged at the initiation of therapy between 2 months and 6 months) diagnosed and genetically confirmed with SMA type I.

We compared changes in motor and respiratory functions between control and study groups, but also between patients receiving the first therapy (T1) and then after receiving two therapies in combination—nusinersen continuing and onasemnogene abeparvovec-xioi (T2). In this respect, for the control group, we evaluated the CHOP-INTEND score at initiation, then every 6 months until 24 months after nusinersen initiation. The same evaluations were performed for the number of ventilation hours and cough assist sessions.

Data analysis was performed in collaboration with the Scientific Research Core from NUCCNNR, using Microsoft Excel and SPSS Statistics for Windows Version 24 and, as the number of patients included in this study was limited, we chose to perform comparisons of the median scores.

## 3. Results

Before adding GT in the study group, no adverse reactions were observed after nusinersen treatment was administered. In the control group, mild agitation (probably due to headache post LP) occurred in two patients, who did not need any antalgic treatments. This might be caused by not respecting the medical recommendation to keep the flat position post-LP.

In the combined therapies group, one patient (Patient 7NZ—see Table 1) died 2 months after GT, after developing serious adverse reactions (the patient had bleeding, respiratory arrest, renal failure needing peritoneal dialysis for one month, was intubated for the whole period after dosing, and was admitted to the Intensive Care Unit), so we noted only the evaluations after the first treatment with nusinersen: the CHOP-INTEND score increased from 8 to 29 points at 6 months after T1 (five doses of nusinersen) and reached 36 points 12 months after T1 (six doses of nusinersen). The score was maintained at 36 points 2 months later, when this patient received GT infusion. There were no changes in ventilation hours and number of cough assist sessions before GT dosing and 1 week later. However, after the onset of GT adverse reactions, the patient was intubated.

After GT, all other patients presented with fever, vomiting, lack of appetite, mild thrombocytopenia and liver enzyme elevation, but only in two patients did the dose of prednisone need to be increased and require prolonged treatment (see Table 1—Patients 1NZ–5NZ and 6ZN). Probably due to a longer (5 months) corticosteroids treatment period, one of the patients presented with demineralization fractures (Figure 1), for which treatment was recommended with supplementary vitamins (C, D) and minerals (calcium, magnesium, phosphorus).

For the only patient who received first onasemnogene abeparvovec-xioi (Patient 6ZN—see Table 2), CHOP increased four points after 6 months from T1 and nine points 6 months after T2 (before the fifth dose of nusinersen). Ventilation time was reduced by 2 h/day for 6 months after T1, while in the same period after T2, no change in NIV hours was needed. For cough assist, a decrease was registered—1 time for T1, but no change after T2.

In the other five patients who received nusinersen first (Patient 1NZ-5NZ—see Table 1), we observed the following:✓Higher improvements in CHOP-INTEND scores 6 months after T1 compared to the same period after T2. Calculated medians for CHOP-INTEND: before T1—15 points, 6 months later it increased to 35 points; before T2—40 points, 6 months later it increased to 52 points.✓Ventilation hours were not modified 6 months after T1, the median being 16 for both evaluations. However, before T2, median for the ventilation hours dropped to 12 and after T2 to 8 h/day.✓Median for cough assist number of sessions was 4 before T1, 3 after 6 months from T1. Before T2, median was 3, and 6 months later it decreased to 1 time per day.

In the control group, for the 6-month period evaluated, we observed slight improvements in ventilation (median decreased by 2 h per day—see Table 2) and insignificant changes in the cough assist session (see Table 2).

Statistical analysis for CHOP-INTEND score evolution 6 months after therapy initiation for both groups—the control group and the study group (at 6 months after T1 and 6 months after T2)—is presented in Table 3.

Among the control group patients, we noticed that the CHOP-INTEND score (see Figure 2) had also slightly improved for the same 6-month period.

For the study group, we observed and compared the two defined time points (T1 and T2). In this respect, the CHOP-INTEND score evolution showed a rapid increase in the first 6 months from T1; then, in the following 6–12 months, the trajectory of the curve grew at a slower pace, as can be seen in Figure 2. The same score progression is observed 6 months later after T2.

## 4. Discussion

Examining the study group’s evolution, we can see that adding the second therapy helps improve ventilation. However, when comparing the results with those of the control group, we observe that ventilation was also ameliorated. It should be stated that these patients respected the respiratory management protocol thoroughly [53], and they performed daily lung volume recruitment [54]. Therefore, we may conclude that, over time, adhering to the standards of care helped improve their respiratory status in conjunction with the child’s growth (including airways), independently of the therapy used. Long-term follow-up and larger study groups could provide more data, so further investigations are needed.

The improved motor function response in patients receiving nusinersen as the first therapy, compared to those who received GT first (despite other studies—see below), could be due to a number of reasons: compliance with the standards of care, individual favorable response of each patient to a specific therapy or age at the time of treatment. In order to investigate other aspects that might provide new information on this matter, we analyzed pNF-H levels [55] from cerebrospinal fluid and plasma—as an indicator of neuronal degeneration, due to disease progression—and correlated them with motor score evolution (ongoing study in NUCCNNR).

We mentioned patients 2N and 6N from the control group who obtained maximum scores on CHOP-INTEND and could not be objectively evaluated. As their motor skills surpass CHOP-INTEND evaluation scores, we believe that their maintaining values do not represent a stagnation. After the 24-month evaluation, these children continued to be assessed with other scales, usually used for SMA II patients.

Regarding nusinersen’s efficacy compared to onasemnogene abeparvovec-xioi, a study from 2019 [56] was conducted, retrospectively, in children aged up to 2 years, and compared two clinical trials for onasemnogene abeparvovec-xioi (AVXS-101-CL-101) and nusinersen (ENDEAR); it analyzed the overall survival rate, the rate of survival without events (event-free survival) and the rate of motor function improvement. This study concluded that GT seems to be more effective than nusinersen in terms of the global survival rate, motor function improvement and need for assisted ventilation. However, the validity of this study has been disputed. For better comparison, further studies are needed, as the inclusion/exclusion criteria and standards of care were not uniform across the groups of patients included in these two studies.

Our study showed that when we include the age of the patients when treatment was initiated and other parameters, treatment efficacy depends heavily on the timing of administration. Sooner is better—ideally, before symptom-onset [57,58], which correlates with very sudden motor regression (in SMA I patients), associated with motor neuron loss (more than 80–90%) [59], a subsequent drop in compound muscle action potential (CMAP) amplitude and active denervation on electromyography [60].

Once this phase is over and the denervation process has stabilized, compensatory changes such as chronic denervation/reinnervation will start. From this moment on, we consider that treatment efficacy is significantly reduced. Therefore, as observed in our patients’ CHOP-INTEND scores, the trajectory curves tend to show slower improvement after the first 6–12 months since the treatment was given, regardless of the type of therapy they received.

To conclude, adding GT after nusinersen does not seem to provide supplementary benefits for motor function or respiratory status, but early treatment results in better outcomes.

## Figures and Tables

**Figure 1 jcm-10-05540-f001:**
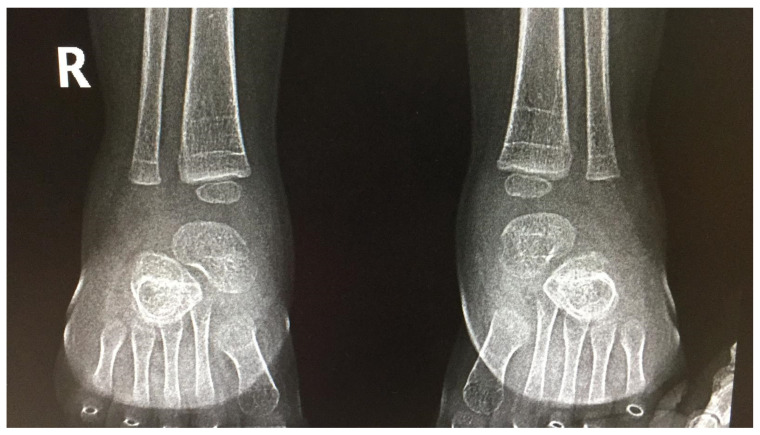
Bilateral foot X-ray (antero-posterior view)—demineralization fractures.

**Figure 2 jcm-10-05540-f002:**
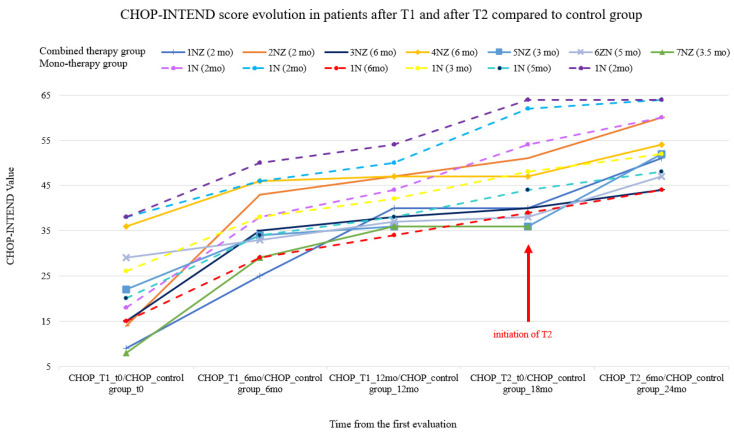
The Children’s Hospital of Philadelphia Infant Test of Neuromuscular Disorders (CHOP-INTEND) score evolution in the control group (dotted lines) and in patients after T1 and after T2 (each patient’s age at treatment initiation appears in parentheses above the graph).

**Table 1 jcm-10-05540-t001:** Clinical and demographic information of study group patients.

Patient	1NZ	2NZ	3NZ	4NZ	5NZ	6ZN	7NZ
Gender	M	M	F	M	M	F	F
Age at symptom onset (months)	1	1	4.5	4.5	2	2	0.5
Age at T1 (months)	2	2	6	6	3	5	3.5
Side effects after T1	no side effects	no side effects	no side effects	no side effects	no side effects	liver enzymes slightly increased; inappetence	no side effects
Age at T2 (months)	9.5	12	12	12	14	23	19
Side effects after T2	AST, ALT higher 2.5-fold from normal; fever	AST, ALT slightly higher; fever; vomit	AST, ALT slightly higher; vomit	AST, ALT slightly higher	AST, ALT higher 3.5-fold from normal; fever	no side effects	severe platelet level drop; AST, ALT higher 10-fold from normal; fever; vomit; bleeding; anemia; respiratory arrest; renal failure
Duration of steroid (months)	3	2	2	2	5	2	2.5
Ventilation hours before T1	16	16	16	0	16	14	16
Ventilation hours 6 months after T1	16	12	16	0	16	12	16
Ventilation hours before T2	14	12	12	0	12	8	16
Ventilation hours 6 months after T2	8	4	10	0	8	8	24
Number of cough assist sessions before T1	6	4	6	3	4	4	6
Number of cough assist sessions 6 months after T1	6	3	4	2	3	3	4
Number of cough assist sessions before T2	4	3	3	2	2	2	3
Number of cough assist sessions 6 months after T2	2	1	2	1	1	2	0
CHOP before T1	9	14	15	36	22	29	8
CHOP 6 months after T1	25	43	35	46	34	33	29
CHOP before T2	40	51	40	47	36	38	36
CHOP 6 months after T2	51	60	44	54	52	47	0

**Table 2 jcm-10-05540-t002:** Clinical and demographic information of control group patients.

Patient	Patient 1	Patient 2	Patient 3	Patient 4	Patient 5	Patient 6
Gender	M	F	M	M	F	F
Age at symptom onset (months)	0.5	1.5	5	3	2	0.5
Age at T1 (months)	2	2	6	5	3	2
Side effects after T1	no side effects	no side effects	mild agitation	mild agitation	no side effects	no side effects
Age at first current study evaluation (months)	14	14	18	16	12	10
Ventilation hours at first current study evaluation	12	10	8	10	12	12
Ventilation hours at second current study evaluation (6 months later)	10	8	6	8	11	12
Number of cough assist sessions at first current study evaluation	4	3	3	4	5	2
Number of cough assist sessions at second current study evaluation (6 months later)	3	3	3	4	4	2
CHOP at first current study evaluation	44	50	34	38	42	54
CHOP at second current study evaluation (6 months later)	54	62	39	44	48	64

**Table 3 jcm-10-05540-t003:** CHOP-INTEND score evolution 6 months after therapy initiation for both groups.

CHOP Evolution		CHOP Evolution in Control Group after 6 Months	CHOP Evolution in Study Group at 6 Months after T1	CHOP Evolution in Study Group at 6 Months after T2
N	Valid	6	6	6
	Missing	0	0	0
Mean		13.33	15.17	9.33
Median		13.00	14.00	9.00
Minimum		8	4.00	4.00
Maximum		20	29.00	16.00

## Data Availability

The data presented in this study are available on request from the corresponding author.

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
