# Peer review of "Combination Therapy with Nusinersen and Onasemnogene Abeparvovec-xioi in Spinal Muscular Atrophy Type I"

_jcm, 2021, doi:10.3390/jcm10235540_

Round 1

Reviewer 1 Report

-I am wondering whether you could superimpose the control group CHOP-INTEND results graph Fig. 2 to the Fig. 3 graph for the combination T1 and T2 therapy cohort. You could use dotted lines for example. It could be as simple as replotting the control group CHOP-INTEND results on Fig. 3 

-by now you must have CHOP-INTEND results for the control group beyond 6 months of treatment. Please present and plot in Figure 2. Did they show stabilization?

Author Response

Thank you for the advice. We completed information on control group CHOP evaluations and we superimposed them, too for the combination therapy cohort. Thus, we obtained one graph with both groups, showing their evolution in time. 

Reviewer 2 Report

In this paper Mirea and co-authors report 7 cases of SMA type I patients treated with the combination therapy of nusinersen and onasemnogene abeparvovecxioi (Zolgensma). They compared the motor function trajectories, ventilation hours and cough assist sessions in the combination therapy treated group to a control group of patients who received nusinersen only (n=6), in order to investigate if combination therapy may be more effective than a single intervention alone. The authors observed that patients who received the combination therapy showed the same motor function trajectory as the monotherapy cohort. They also observed that the evolution of motor function was better in the 6 months following the first therapy than in the first 6 months after adding the second treatment and concluded that an early treatment is more important than a combined treatment.

As nusinersen and zolgensma are new treatments and only become available for SMA patients in the recent years. This report has hence delivered important information in this field regarding different potential therapeutic regimens. However, there are still a few issues need to be addressed in the current study.  

Major comments:

  • In this study, the authors set patients who received nusinersen only as the control group. If possible, it would be great to have another control group with patients received Zolgensma only.
  • It would be really great if the authors may include any data they may have on pNF-H, creatinine or Troponin-I levels as they mentioned in the ‘Materials and Methods’ (page 4, line 156-158). It would be interesting to know if there is any difference in biomarkers at the molecular levels between the monotreatment and combined treatment, to further validate the conclusion that no difference was observed in the clinical motor function measures between monotreatment and combined treatment groups.
  • In Figure 2, the authors only showed the changes of CHOP values in the first 6 months. Can author also show one more figure of CHOP values in the control group at more time points similar to the time points used in Figure 3 in the combined treatment group?
  • I cannot see any results on statistical analysis in either figures or tables, although it says on page 4 line 180-182 that they performed data analysis with Excel and SPSS using medians comparisons of the median scores.
  • The author may add some acknowledgement of study limitations as: 1) this study is based on a small cohort of only 7 patients in the combined treatment group and only 6 patients in the nusinersen-only treatment group; 2) this study only measured motor functions in 6 months after the combined treatment and therefore only reflect the short-term effect. The effects in long-term between the combined-treatment and mono-treatment groups are still unknown.   

Minor comments:

  1. Page 1, ‘Abstract’- line 20: either remove ‘However’ or rephrase that sentence.
  2. Page 10 line 289-291 at the end of the Discussion session the author mentioned that ‘In the meantime, for a better quality of life, we should consider the use of mobile mechatronic/ robotic orthotic devices, both for the upper and for the lower limbs for patients who receive treatments long after disease onset’. This sentence seems quite sudden as no previous mention of mobile mechatronic/ robotic orthotic devices in this study. Not sure how relevant this sentence is to the entire study.

Author Response

Thank you very much for the review. 

Unfortunately, we haven't any naive Zolgensma patients. So we can't have a Zolgensma group pacients in our study.

We eliminated data about biological markers, as these will constitute the object of another's trial, that is ongoing in our hospital and we can't yet provide data. We observed patients from this point of view, but we won't use the information yet, as we have to dig deeper to find out why pNF-H raise (even 10 times for some patients) after Zolgensma dosing and than they decrease in time... As these evaluations weren't the subject of this present paper, we decided to eliminate completely the mentionning of them.

We also added the information for CHOP scores evolution in control group and we superimposed it for the combination therapy cohort in order to be better observed the evolution of all treated patients, for the same perriod of months from starting therapy.

Your comments for adding suplementary aknowledgements for study limitations were very usefull, too and we have added that, too. 

We revised also line 20 from abstract. And also... we eliminated the sentence from page 10 (lines 289-291) on the mobile mechatronic/ robotic orthotic devices.

Thank you once again for great advices.